# How Is Work–Life Balance Arrangement Associated with Organisational Performance? A Meta-Analysis

**DOI:** 10.3390/ijerph17124446

**Published:** 2020-06-21

**Authors:** Kapo Wong, Alan H. S. Chan, Pei-Lee Teh

**Affiliations:** 1Department of Systems Engineering and Engineering Management, City University of Hong Kong, Kowloon, Hong Kong; alan.chan@cityu.edu.hk; 2School of Business, Gerontechnology Laboratory, Monash University Malaysia, Bandar Sunway 47500, Malaysia; teh.pei.lee@monash.edu

**Keywords:** organisational performance, organisational commitment, productivity, meta-analysis

## Abstract

The impacts of the work–life balance arrangement on organisational performance is a growing concern amongst researchers and practitioners. This study synthesised 202 records from 58 published papers to evaluate the relationship between the work–life balance arrangement and organisational performance by means of a meta-analysis. The organisational performance was measured based on six perspectives, including career motivation, employee attendance, employee recruitment, employee retention, organisational commitment, and productivity. The results showed a positive relationship between the work–life balance arrangement and organisational performance (OR: 1.181, 95% CI: 1.125–1.240, *p* < 0.001). Of the six perspectives, only career motivation, employee attendance, employee recruitment, and employee retention were significantly associated with the work–life balance arrangement. The moderators affecting the relationship between the work–life balance arrangement and organisational performance were gender, sector, and employee hierarchy. The results provide theoretical suggestions on the effectiveness of the work–life balance arrangement in terms of the six perspectives related to organisational performance.

## 1. Introduction

Work–life balance is a highly discussed issue worldwide. A healthy work–life balance not only tremendously affects the physical and mental development of an individual but also the sustainability of organisations. Therefore, a work–life balance arrangement (WLBA) should be implemented to safeguard well-being and improve the performance of workers. WLBA refers to policies or practices that assist workers to maintain a healthy work–life balance [1]. Most workers place WLBA in a higher position than wages. In the United Kingdom, a survey interviewing 2000 workers conducted by Scott [2] in 2016 has revealed that more than 50% of the respondents look for jobs that promote work–life balance rather than those with an attractive salary and benefits. In the United States, FlexJobs [3] has conducted a survey which indicates that approximately 80% of the respondents rank work–life balance higher than a better salary when searching for jobs amongst 1100 working parents. WLBAs have different types, including family friendly policies, flexible work hours or schedules, incentive programs, workplace health programs, and work–life balance programs [4,5,6]. Furthermore, organisational performance (OP) can be evaluated in different perspectives, including career motivation (CM), employee attendance (EA), employee recruitment (EC), employee retention (ET), organisational commitment (OC), and productivity (PR) [7,8]. These five common types of WLBAs and the six perspectives on OP were used for keyword searching in the data extraction part of this meta-analysis. The features of each type of WLBA and the six perspectives on OP are discussed below. The association of WLBAs with these six perspectives on OP was explored in this study.

### 1.1. Work–Life Balance Arrangement

#### 1.1.1. Family-Friendly Policies

Family-friendly policies are a series of complementary benefits, including childcare, maternity leave, and parental leave [9]. These policies aim to retain talented workers by resolving their conflict between work and their personal life [10,11]. Several studies have found that family friendly policies have a positive impact on OP [6,12]. A meta-analysis conducted by McNall et al. [13] has demonstrated that family friendly policies improve OP. However, Durst [10] has claimed a negative association between family friendly policies and OP. The results of the relationship between family friendly policies and OP are still inconsistent. 

#### 1.1.2. Flexible Working Hours

Flexible working hours allow workers to adjust their work schedule to meet personal and work needs [4]. Several studies have suggested that flexible working hours improve OP [14,15], whereas some studies have revealed a negative relationship between flexible work schedules and OP [16,17]. Thus, previous studies have reported inconsistent findings on the relationship between flexible working hours and OP.

#### 1.1.3. Incentive Program

An incentive program, either monetary or nonmonetary, recognises workers who achieve organisational goals [18]. The implementation of an incentive program improves the OP of marketing managers [19], juvenile detention workers [20], and information centre employees [21]. Several studies have found that workers with a high level of autonomy perform better [22,23]. Conversely, the study of Al et al. [14] investigating salespeople in Istanbul has reported no relationship between incentive programs and OP. Thus, findings on the effects of incentive programs on OP have been inconclusive. 

#### 1.1.4. Workplace Health Program

A workplace health program is a comprehensive set of health promotion and protection strategies that are linked to communities for safeguarding the safety and health of workers [24,25]. The measures used by the program include health insurance for both workers and their dependents, discounted fitness programs, and mental health services [5]. Some studies have indicated that workplace health programs have a positive impact on OP [25,26]. For instance, Shephard [26] has found that providing life insurance to employees improves OP. The literature has reported consistent findings showing that workplace health programs are beneficial to OP.

#### 1.1.5. Work–Life Balance Program

A work–life balance program is the combination of several work–life balance initiatives [8,27]. Meyer et al. [28] has found that implementing work–life balance programs for working mothers can improve OP. The effectiveness of the implementation of this program on the OP has been explored in the United States, Japan, and Australia [29,30,31].

### 1.2. Organisational Performance 

#### 1.2.1. Career Motivation (CM)

Career motivation is the desire to exert effort to achieve career goals [32,33]. Some studies have found that CM is boosted by WLBA [34,35,36]. Aluko [34] has shown that African female employees working in academia and banking sectors improve their CM under WLBA. Williams et al. [36] have demonstrated that the CM of managers working in North America is stimulated by WLBA. Unlike Aluko and Williams et al., Aronsson et al. [37] have argued that employee motivation is negatively associated with supervisor support. The findings on the effects of WLBA on CM are inconsistent.

#### 1.2.2. Employee Attendance (EA)

Employee attendance refers to employees being present for work. Chow et al. [38] have revealed that the EA of service industries in Singapore is ameliorated by WLBA. Halpern [39] has demonstrated that WLBA enhances the EA in the United States. Nevertheless, Lidwall et al. [40] have claimed that WLBA increases the chance of long-term sickness absence. The findings for WLBA and EA are not straightforward. 

#### 1.2.3. Employee Recruitment (EC)

Employee recruitment refers to the process of seeking potential and qualified people for organisations [41]. Some studies have revealed that an organisation with a work–life balance initiative increases its competitiveness in recruitment [41,42,43,44]. The implementation of WLBA in biotechnology firms has attracted potential job candidates [41]. Hospitality and tourism industries in Australia that implement WLBA have a relatively higher chance of recruiting promising applicants [42]. Ehrhart et al. [43] have demonstrated that organisations in the United States with a WLBA have better recruitment than those without this arrangement. The results for WLBA and EC are straightforward.

#### 1.2.4. Employee Retention (ET)

Employee retention means that organisations provide a supportive working environment to retain their workers [42]. Some studies have suggested that WLBA increases ET [45,46,47,48]. Houston and Waumsley [47] have indicated that WLBA enhances the ET for male workers in the United Kingdom. Additionally, ET has had positive effects on companies, such as a decrease in financial costs, a reduction in accident rates, and an increase in the quality of work [8,49,50,51]. Previous literature has reported consistent results on WLBA and ET. 

#### 1.2.5. Organisational Commitment (OC)

Organisational commitment is the acceptance of the organisational values, willingness to make an effort, and the desire to continue employment in companies [52]. OP can be enhanced by the engagement of employees [52]. Nurses in Turkey and female construction workers in the United States have a high OC while implementing WLBA in their organisations [53,54]. Conversely, Riaz et al. [55] have argued that WLBA has negative impacts on the OC of investors in Pakistan. Allen et al. [49] have found a negative correlation between incentive programs and OC. Thus, inconsistent findings on WLBA and OC have been reported. 

#### 1.2.6. Productivity (PR)

Productivity refers to the capability of producing good service [56]. The higher the PR of employees is, the better the OP will be [33]. A considerable amount of research has indicated that WLBA increases PR [57,58,59,60]. Positive relationships between the promotion of WLBA and the PR of female workers in South Africa [58] and financial employees in the United Kingdom [60] have been found. However, Bloom et al. [57] have claimed that WLBA in manufacturing workers in the United States, France, Germany, and the United Kingdom have negative relationships with PR. Therefore, the findings for WLBA and PR are not consistent.

### 1.3. Relationship between WLBA and OP

The relationship between WLBA and OP can be described based on social exchange theory [61], which implies an evaluation of cost and rewards. Caillier [62] claimed that incentives can be offered to employees in return for devotion to an organisation. The employers formulate favourable policies and provide support to motivate employees to contribute more, which results in improved productivity, attendance, and retention [63]. Giovanis [64] used expectancy theory to depict the nexus between WLBA and OP, showing that the provision of WLBA allows more resources for employees who are inclined to perform better. Boundary theory was suggested to assess how an individual separates or integrates the work and personal life domains to achieve a balance [64,65]. One of the WLBAs, flexible working arrangements, allows higher job autonomy for employees. Hence, employees have better management on each role and are likely to contribute more to work.

Numerous recent studies have identified the salutary effects of WLBA on OP, including enhancing retention, increasing profit, alleviating attendance, sustaining commitment, and stimulating motivation [63,65]. However, some studies have stated that WLBA has no effect or even has deleterious impacts on OP (e.g., CM, EA, OC, and PR). Meanwhile, several researchers have demonstrated that reduction in the effectiveness of work–life balance policies in an organisation can enhance the work–life balance [66,67]. Therefore, no consistent conclusion can be drawn for the relationship between WLBA and OP. To address this issue, the relationship between WLBS and the six perspectives (CM, EA, EC, ET, OC, and PR) related to OP was estimated with the use of a meta-analysis approach.

### 1.4. Aims 

The inconsistent findings between WLBA and OP (CM, EA, EC, ET, OC, and PR) are demonstrated above. The effectiveness of WLBA, though having been employed in most organisations, is not completely verified. To our knowledge, no meta-analysis has been conducted on the relationship between WLBA and OP. In addition, OP severely affects the decisions of human resource management as well as the long-term development of organisations. Therefore, in no way can the importance of OP be overlooked, particularly under the establishment of WLBA. Considering the inconsistent findings on the relationship between WLBA and OP and the lack of research on it, a meta-analysis must be conducted to investigate the relationship between WLBA and OP. The findings of this study can provide a profound understanding of the relationship between WLBA and OP for the practitioners of human resource management. Such understanding can help to evaluate the inadequacy of the current policies on the basis of the results of six perspectives (CM, EA, EC, ET, OC, and PR) on OP and adjust policies to increase performance. 

## 2. Materials and Methods

The relationship between WLBA and OP was investigated with a meta-analysis in this study. First, a literature search, which is the process of extracting potential and qualified articles based on predetermined criteria, was carried out. Second, coding—the extraction and classification of information from selected papers—was conducted. Finally, the effect sizes and publication bias were analysed. 

### 2.1. Literature Search

The studies for the literature search were extracted from five electronic databases—namely, Google Scholar, Science Direct, ProQuest, MEDLINE, and PubMed. The keywords for browsing the studies were “work–life balance OR workplace health programs OR incentive programs OR work–life balance programs OR family-friendly policies OR flexible working hours OR flexible work schedules” AND “organisational performance OR employee retention OR recruitment OR employee attendance OR career motivation OR productivity OR organisational commitment”. The criteria of paper extraction included: (1) studies of the investigation, focusing on the association between WLBA and OP; (2) the provision of sufficient data for the computation of odds ratios; (3) articles published in English; and (4) studies containing original data. Some articles were excluded from the paper selection, including letters, case reports, comments, duplicated data from other studies, editorials, and systemic reviews. 

### 2.2. Coding 

Information from the studies—including authors, gender, publication year, average age, origin, sector, employee hierarchy, WLBA, and OP—was extracted. The average age of the participants was coded into two categories: less than 50 years old and 50 years old and above. Fifty years old was selected as the cut-off age, because numerous researchers had defined older workers as those aged 50 years old and above [68,69]. The sector was coded into three categories: healthcare, manufacturing, and cross-sector. If the studies investigated more than one sector without classification, cross-sector was applied for these studies. For example, the study of Aronsson [37] involved the agriculture, transport, and food sectors. Thus, the study was categorised as cross-sector. Employee hierarchy was coded into three categories, including general staff (blue-collar, administrative, support, and service employees), managers, and mixed employees (studies combining general staff and managers). WLBA had six categories, including workplace health programs, incentive programs, work–life balance programs, family-friendly policies, flexible working hours, and flexible work schedules. OP constituted six categories: ET, EC, CM, EA, PR, and OC. 

### 2.3. Meta-Analysis

Statistical analyses were performed by using Comprehensive Meta-Analysis Software, version 3.0. The random effects model was used in the analysis due to differences in treatment effects among individual studies. Additionally, the trim-and-fill method was used to test and adjust the publication bias [70,71]. A funnel plot was a scatter plot of the effect size against the sample size of the study [72]. The presence of publication bias resulted in an asymmetrical funnel plot. I-squared (I2) statistics and Q were indicators of heterogeneity. I2 was independent of the number of articles and the scales of effect size, and Q was sensitive to the number of articles [73]. Therefore, only I2 was used for determining the heterogeneity. The greater the I2 static value was, the higher the heterogeneity would be [68]. Furthermore, a subgroup analysis was conducted to investigate the moderating effects and possible sources of heterogeneity among the selected articles.

## 3. Results

A total of 1852 potentially relevant articles were identified from the electronic database. Then, 1730 articles were excluded on the basis of the selection criteria. Subsequently, 64 articles were excluded due to shared identical population (*n* = 2), systematic review (*n* = 4), no available variables (*n* = 34), and insufficient data for the computation of the odds ratio (OR) (*n* = 24). A total of 202 records from 58 articles were retained for the meta-analysis, as shown in Figure 1. 

### 3.1. Characteristics of Included Studies

The 58 articles had 1,478,798 participants. The greatest sample size was 74,952 participants [74], while the smallest sample size was 80 participants [75]. More than half (58.4%) of the studies contained female and male employees. In 22.3% of the articles, only female workers were present, while 18.8% of the articles contained only male workers. One article did not mention the gender. Out of the studies, 47.5% were published from 1998 to 2007, and 52.5% of them were published from 2008 to 2018. Among the studies, 90.6% participants were aged below 50, and 5.4% participants were aged 50 and above. Eight articles did not report the age of the participants. The majority (81.2%) of the studies were conducted in Europe. The remaining studies were carried out in Asia (8.4%), North America (6.9%), and Australia (3.5%). The characteristics of the included studies are summarised in Table 1.

### 3.2. Overall Effect Size Based on Random Effect

The aggregated odds ratio for the relationship between WLBA and OP was 1.181 (95% confidence interval (CI): 1.125–1.240, *p* < 0.001). WLBA was significantly associated with ET (OR: 1.357; 95% CI: 1.180–1.561; *p* < 0.001), EC (OR: 1.321; 95% CI: 1.119–1.561; *p* = 0.001), CA (OR: 1.283; 95% CI: 1.084–1.519; *p* = 0.004), and EA (OR: 1.195; 95% CI: 1.124–1.271; *p* < 0.001). WLBA was non-significantly associated with PR (OR: 1.139; 95% CI: 0.961–1.351; *p* = 0.134) and OC (OR: 1.046; 95% CI: 0.835–1.309; *p* = 0.696). The forest plots of aggregated studies grouped by OP are illustrated in Figure 2a–f. 

### 3.3. Publication Bias

The trim-and-fill method was used to test and adjust the publication bias in the meta-analysis and estimate the number of “missing” or unpublished studies through testing for the asymmetry of the funnel plot [70]. New data points were added to the adjustment of publication bias for CM (*n* = 4, OR: 1.422; 95% CI: 1.215–1.664), EA (*n* = 23, OR: 1.046; 95% CI: 0.978–1.101), ET (*n* = 4, OR: 1.481; 95% CI: 1.278–1.716), and PR (*n* = 11, OR: 0.899; 95% CI: 0.756–1.069). No new data points were added to the adjustment of the publication bias for EC and OC. 

### 3.4. Test of Heterogeneity

The I^2^ static value was the indicator of heterogeneity. This value was independent of the numbers of articles and the scales of effect size [73]. The I^2^ static values for CM, EA, EC, ET, OC, and PR were 91.395%, 91.653%, 93.248%, 69.027%, 94.232%, and 93.506%, respectively, which indicated a considerably high heterogeneity among the six perspectives on OP. Table 2 shows the heterogeneity between WLBA and the six perspectives of OP. To determine the potential sources of heterogeneity, the moderating effects of variables were examined. 

### 3.5. Moderator Analysis

The moderating effects were investigated using a subgroup analysis for age, gender, publication year, origin of study, sector, and employee hierarchy. All the moderator variables were categorical variables.

Gender had a significant moderating effect (*p* = 0.008). Males (OR: 1.270; 95% CI: 1.167–1.383) and females (OR: 1.248; 95% CI: 1.154–1.349) shared a similar association. The sector showed a significant moderating effect (*p* = 0.031). The manufacturing sector (OR: 1.406; 95% CI: 1.180–1.676) showed a stronger association than the healthcare sector (OR: 1.224; 95% CI: 1.101–1.360). Employee hierarchy demonstrated a significant moderating effect (*p* < 0.001). Managers (OR: 1.430; 95% CI: 1.273–1.607) demonstrated a stronger association than the general staff (OR: 1.152; 95% CI: 1.086–1.222). The subgroup analysis illustrated that the publication year (*p* = 0.156), age (*p* = 0.099), and origin of the study (*p* = 0.381) had non-significant moderating effects. Table 3 shows the results of the moderators using a subgroup analysis.

## 4. Discussion

A total of 202 records from 58 articles were analysed for the relationship between WLBA and OP. The results indicated that WLBA was able to improve the OP. Additionally, gender, sector, and employee hierarchy showed significant moderating effects on the relationship between WLBA and OP. 

### 4.1. The Relationship between WLBA and OP

WLBA is significantly associated with CM, EA, EC, and ET. However, WLBA is not significantly associated with OC and PR. CM has been found to be significantly associated with WLBA. WLBA allows employees to have a high autonomy and flexibility at work, which can motivate the performance of employees [76,77]. This finding substantiates the previous studies of Aluko [34], Bui et al. [35], and Williams et al. [36], which have demonstrated that CM can be boosted by WLBA.

WLBA is significantly associated with a high EA. WLBA allows employees to compensate for times when they were on leave. For instance, employees are allowed to work under a flexible work schedule to compensate for the time used for dealing with family matters or other sudden personal affairs during working hours [39,78]. This workplace flexibility can improve the attendance of the employees. This finding confirms the previous studies of Baltes et al. [79] and Chow and Chew [38], which have suggested a positive relationship between an appropriate WLBA and EA.

WLBA is significantly associated with EC. WLBA expands the recruitment of potential and qualified talents, such as applicants with children or dependents. These types of workers can be arranged to work under a flexible work schedule [80]. This result is consistent with the previous studies of Dex and Scheibl [81] and Eaton [41], which have stated that the work–life balance plays an important role in the decision-making process of the recruitment of employees. Moreover, financial incentive is no longer the most significant consideration among employees. Therefore, WLBA can affect the EC. 

ET has been found to be significantly associated with WLBA. Providing training programs and flexible working hours are two of strategies of WLBA. Workers are able to acquire new skills and knowledge through training programs, and the employees can perceive potential career growth within the organisations [45,47]. Flexible working hours can increase the flexibility for employees to cope with nonwork affairs during working hours [82]. Thus, employees have a high tendency to work in the company if the organisation provides favourable work–life balance policies for employees. This result is in line with the previous studies of Yu [46] and Halpern [39], which have demonstrated that assisting employees to achieve a work–life balance can reduce employee turnover.

WLBA is not significantly associated with OC. OC may not be developed by the implementation of WLBA. Camilleri [83] has found that the major factors significantly influencing employee commitment are personality traits, work position, and educational level. Further studies are recommended to investigate the insignificant relationship between WLBA and OC.

WLBA is not significantly associated with PR, possibly due to the ambiguous boundary between work and nonwork domains. One of the initiatives of work–life balance, flexible work schedules, may generate an unclear declination between work and nonwork domains. Some employers may require employees to work additional hours by taking advantage of flexible work schedules. The misuse of flexible work schedules may result in a low PR. Bloom et al. [56] and White et al. [59] have demonstrated a negative relationship between WLBA and PR. However, several studies have found that WLBA has a positive relationship with PR [58,60,84,85].

In a nutshell, there is a positive relationship between WLBA and OP. Practically, WLBA aims to mitigate the conflict between work and nonwork roles that the well-being of employees can be maintained [65]. Employees with a good physical and mental health have better work performances [86]. WLBA creates mutual benefits between employees and organisations. Importantly, tailoring work-life balance practices to meet the needs of employees is an ideal approach. Apart from WLBAs, other factors might influence the implementation of WLBAs and OP. The cooperation of co-workers might affect the working attitudes of employees to some extent [86]. The sufficiency of financial assistance from the government and the internal budget might affect decisions on the execution of policies [87]. An appropriate and safe workplace design might enhance the performance of employees [88]. Furthermore, the study of Li et al. [89] revealed that the company ownership structure impacted the corporate social performance of an organisation. The regulatory environment and corporate governance mechanisms might considerably influence the sustainability and corporate social performance [90]. Various factors might possibly influence the relationship between WLBA and OP. Current studies mainly adopted these factors as moderators when investigating the association between WLBA and OP (e.g., [86,91]).

### 4.2. Moderating Variables

Gender has a significant moderating effect on the relationship between WLBA and OP. Males have a slightly better OP than females under WLBA, which is possibly due to different work attitudes and capabilities between males and females. Male workers are goal-oriented and female workers are cooperative and willing to communicate with others [92]. However, both male and female workers have the same goal, which is to resolve problems and enhance business efficiency [92,93]. Therefore, a slight difference has been observed between males and females on the relationship between WLBA and OP.

Sector has a significant influence on the overall effect size. The manufacturing sector has a stronger association than the healthcare sector. The main reason may be that automated machines are applied in the manufacturing sector. These machines assist in accelerating the manufacturing process, increasing the PR and reducing the input of the labour force [94]. However, workers in the healthcare sector normally have to handle tasks in person. Therefore, the manufacturing sector may have a relatively higher PR than the healthcare sector.

Employee hierarchy demonstrates a significant moderating effect on the relationship between WLBA and OP. Managers have a better OP than general employees under WLBA. Managers have a higher position than general employees; thus, managers may take greater responsibilities than general employees [95,96,97]. General employees tend to follow the orders and directions of the managers and bosses [98]. Therefore, managers have a stronger association in the relationship between WLBA and OP. 

Publication year and age have no significant effect on the overall effect size. The results for publication year are consistent with the studies of White et al. [59], which have indicated no relationship between WLBA and OP. For age, previous studies have also indicated an inconsistent finding on the relationship between WLBA and OP. Silverstein [99] has shown that older employees usually have a low PR in the organisation. Meanwhile, Appelbaum et al. [100] have claimed that older workers can have high PR due to seasoned experience and interpersonal skills. Additionally, the country of origin has no significant influence on the overall effect size. 

### 4.3. Significance of the Study 

The study extends our knowledge of the influences of WLBAs on different perspectives on OP, in which only CM, EA, EC, and ET were significantly associated with WLBAs. Surprisingly, no significant association of WLBAs with OC and PR was found. These findings have important implications for the optimisation of the current WLBA. Identifying possible ways to heighten the OC and PR of employees is an significant step for the formulation of WLBAs to be adjusted to meet the needs of employees as well as enhance the OP. Further empirical study with more focus on the effects of WLBA on OC and PR is recommended. Furthermore, the study revealed that gender, sector, and employee hierarchy have significant effects on the association between WLBA and OP. These influential factors and other third factors discussed above should therefore be considered in developing a thoughtful WLBA. The finding offers research insights into how WLBAs may enhance OP and the inadequacy of current WLBA in contributing to OP. 

### 4.4. Theoretical Implication

The impacts of work–life balance policies on the OP were evaluated by the computation of effect sizes. In this study, OP was investigated in six perspectives, four of which had a significant relationship with WLBA—namely, CM, EA, ET, and EC. WLBA was found to have no significant relationship with PR and OC. WLBA was able to retain workers, improve their attendance, and attract new people. Workers aspired to achieve targets in their career fields under the establishment of WLBA. However, the outputs of the organisations and the employee loyalty towards companies obtained from WLBA were not beneficial. This finding implied that WLBA only changed the attitudes and behaviours of the workers in terms of self-development rather than the organisations. In addition, several variables influenced the relationship between OP and WLBA. Firstly, males and females had distinct priorities in life. Thus, gender served as a moderator in the relationship between OP and WLBA. Secondly, different effect sizes in the relationship between OP and WLBA amongst disparate sectors were due to the heterogeneous natures of different sectors. Lastly, employee hierarchy influenced the relationship between OP and WLBA because of the different duties amongst positions.

### 4.5. Practical Implication

Assisting employees to achieve work–life balance is one of the important duties of companies. The findings have shown that WLBA is significantly associated with the improvement on OP. Employers may introduce some flexible work initiatives, such as teleworking, home-working, and e-working, to help employees achieve a balance between work and personal life [27]. Furthermore, adequate breaks can be provided for employees on work days to retain talented staff by assisting them in striking a balance between work and life [42]. In addition, organisations should provide training programs for managing and dealing with the needs of employees. Thus, a work–life balance culture can be cultivated in the organisations through management [13]. Job nature is also a crucial factor to be discussed. Several researchers have stated that a high level of job autonomy, the provision of additional resources, and psychological rewards are associated with a healthier work–life balance [101,102,103]. Most importantly, manpower is regarded as a competitive advantage to any company. Hence, employers should regularly communicate with employees to understand their circumstances and try to accommodate the needs of employees in work- and life-related aspects, which can enhance the loyalty and PR of employees [104]. 

### 4.6. Limitations

Several limitations were found in this study. Firstly, only articles written in English were collected for this meta-analysis. A small number of published articles written in other languages were excluded. This meta-analysis consisted of a large number of studies. The validity and reliability of this meta-analysis could be generalised. Secondly, the effect sizes of different studies were adjusted by different controlled variables, such as gender, educational level, and marital status. Therefore, investigating the effect sizes adjusted by the same types of controlled variables, if possible, might be an accurate approach. Thirdly, participants in the records were mostly from the healthcare and manufacturing sectors. Additional sectors should be investigated to have a comprehensive evaluation on the association between WLBA and OP. 

## 5. Conclusions

This study used a meta-analysis to summarise a total of 202 records from 58 studies on the association between WLBA and OP. OP was investigated based on six perspectives—CA, EA, EC, ET, PR, and OC. A positive relationship between WLBA and OP was found. Moreover, only CM, EA, EC, and ET were significantly associated with WLBA. Gender, sector, and employee hierarchy were the significant moderators in the relationship between WLBA and OP. These findings enhance our understanding of the influential level of WLBAs on different perspectives on OP. Various third variables might influence the relationship between WLBA and OP, and thus these third variables are suggested to be moderators in the further investigation of the relationship between WLBA and OP. Furthermore, social exchange theory and expectancy theory applied to the relationship between WLBA and OP have a similar proposition that favourable policies might likely to motivate workers to make greater contributions towards organisations. Boundary theory can be used to explore the psychological states of workers in confronting a vague role boundary. Another further research regarding how WLBA might influence the psychological well-being would be worthwhile to conduct. The findings of this study provided recognition on the impacts of WLBA on CM, EA, EC, ET, PR, and OC, which served as a reference for organisations to modify or adjust the WLBA to boost PR and OC. Employee-driven work–life balance policies, which have to be established to effectively reap the profits generated by labour, would be beneficial to organisations in the long run. This updated meta-analysis on WLBA and OP was conducted in the hope of inspiring future research to conduct empirical studies on WLBA and other perspectives related to OP.

## Figures and Tables

**Figure 1 ijerph-17-04446-f001:**
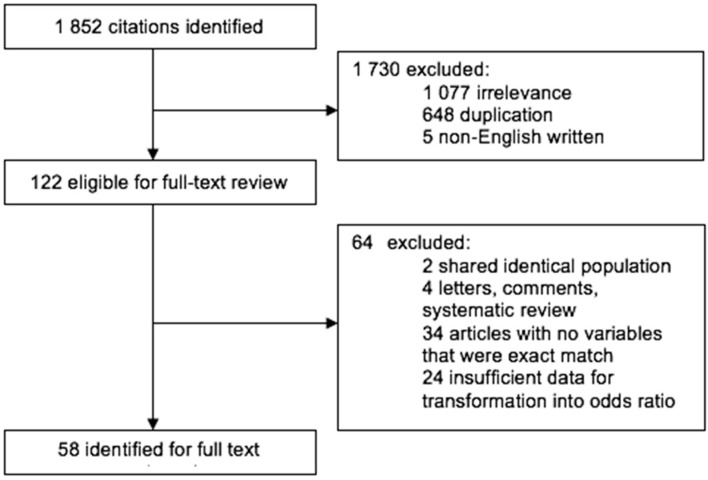
Flow diagram for identifying relevant studies. Note: OR: odds ratio; CI: confidence interval.

**Figure 2 ijerph-17-04446-f002:**
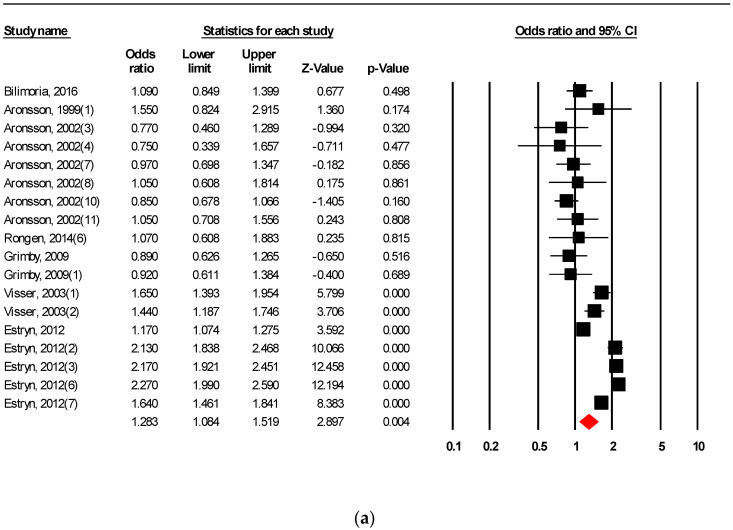
Forest plot of (**a**) career motivation (CM), (**b**) employee attendance (EA), (**c**) employee recruitment (EC), (**d**) employee retention (ET), (**e**) organisational commitment (OC), and (**f**) productivity (PR). Note: In this forest plot, the midpoint of a square provides the odds ratio estimated for the study. The area of the square is proportional to the weight of the study in the meta-analysis. The width of the horizontal line shows the confidence interval. The centre of the diamond represents the overall odds ratio, and its width refers to the confidence interval.

**Table 1 ijerph-17-04446-t001:** Characteristics of the included studies.

Characteristics	Number of Records	Proportion
**Gender**		
Both female and male	118	58.4%
Female	45	22.3%
Male	38	18.8%
Not mentioned	1	0.5%
**Publication year**		
1998–2007	96	47.5%
2008–2018	106	52.5%
**Average age**		
<50 years old	183	90.6%
≥50 years old	11	5.4%
Not mentioned	8	4%
**Origin**		
Europe	164	81.2%
Asia	17	8.4%
North America	14	6.9%
Australia	7	3.5%

**Table 2 ijerph-17-04446-t002:** Random effects sizes between work–life balance arrangement (WLBA) and organisational performance (OP) (CM, EA, EC, ET, OC, and PR).

		Effect Size and 95% CI		Heterogeneity		Adjustment for Publication Bias		
	Number of Records	OR	95% CI	*p*-Value	I^2^ (%)	Imputed Point	New OR	95% CI
**CM**	18	1.283	1.083–1.519	0.000	91.395	4	1.422	1.215–1.664
**EA**	91	1.195	0.000–1.271	0.000	91.653	23	1.038	0.978–1.101
**EC**	8	1.321	1.119–1.561	0.000	93.248	0	1.321	1.119–1.561
**ET**	17	1.357	1.180–1.561	0.000	69.027	4	1.481	1.278–1.716
**OC**	23	1.046	0.835–1.309	0.000	94.232	0	1.046	0.835–1.309
**PR**	45	1.139	0.961–1.351	0.000	93.506	11	0.899	0.756–1.069
**Overall**	202	1.181	1.125–1.240	0.000	93.597			

**Table 3 ijerph-17-04446-t003:** Results of the moderators using a subgroup analysis.

Moderators	Odds Ratio	95% CI Lower Limit	95% CI Upper Limit	*p*-Value
**Gender**				
Overall	1.206	1.154	1.259	0.008
Males	1.270	1.167	1.383	<0.001
Females	1.248	1.154	1.349	<0.001
**Sector**				
Overall	1.161	1.107	1.218	0.031
Cross-sector	1.122	1.061	1.186	<0.001
Manufacturing	1.406	1.180	1.676	<0.001
Healthcare	1.224	1.101	1.360	<0.001
**Employee hierarchy**				
Overall	1.168	1.114	1.225	<0.001
Mixed employees	1.015	0.907	1.137	0.792
Managers	1.430	1.273	1.607	<0.001
General staff	1.152	1.086	1.222	<0.001
**Publication year**				
Overall	1.190	1.133	1.251	0.156
1998–2007	1.226	1.150	1.306	<0.001
2008–2018	1.139	1.054	1.232	0.001
**Average age**				
Overall	1.198	1.141	1.258	0.099
≥50 years old	1.198	1.140	1.259	<0.001
<50 years old	0.870	0.585	1.293	0.490
Not mentioned	1.551	1.096	2.195	0.013
**Origin**				
Overall	1.196	1.138	1.258	0.381
Asia	1.297	1.124	1.496	<0.001
Europe	1.198	1.129	1.271	<0.001
Australia	1.137	0.995	1.299	0.058
North America	0.967	0.671	1.395	0.859

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
