# Peer review of "How Is Work–Life Balance Arrangement Associated with Organisational Performance? A Meta-Analysis"

_ijerph, 2020, doi:10.3390/ijerph17124446_

Round 1

Reviewer 1 Report

This is an excellent Meta-Analysisand well-performed by the authors. The topic of study is of high importance related to work–life balance arrangement associated with organisational performance. As a reader it was enjoyable to read the article and as a reviewer I could not suggest any modifications in the content.

Reviewer 2 Report

This paper presents a Meta-Analysis of 202 records from 58 published papers to evaluate the relationship between work-life balance arrangement and organisational performance. The authors analyse the relationship on six perspectives, and after the literature search and the extraction and classification of information of the selected papers, they also analyse effect sizes and publication bias.

The research question is clear and well defined, the topic is quite original and the paper is accurate. Moreover, it contains a good literature review concerning the topic. The discussion of results is sound and well-conducted, and there is a clear description of the limitations and future research opportunities. I think that the conclusions are interesting for the readership of the Journal and that the paper will attract a wide readership. The paper does read fluidly, and it is written in Standard English.

For those reasons, I think there is an overall benefit to publishing this work.

  • Does the introduction provide sufficient background and include all relevant references?

The introduction clearly presents the research question, and all the paper is good.

  • Is the research design appropriate?

The research on which the paper is based seems well designed. The analysis proposed follows a well-known method in quantitative research.

  • Are the methods adequately described?

Methods are correctly described.

  • Are the results clearly presented?

Results are clearly presented.

  • Are the conclusions supported by the results?

The conclusions are drawn appropriately based on the data presented.

Review Comments to the Authors

Dear Authors, congratulations on submitting a welcome contribution to this field of study. I really appreciated reading your paper and I think it should be published.

Thank you!

Reviewer 3 Report

attached

Round 2

Reviewer 3 Report

Congrats on a successful revision.